# Scarpa Fascia Preservation to Reduce Seroma Rate on Massive Weight Loss Patients Undergoing Abdominoplasty: A Comparative Study

**DOI:** 10.3390/jcm12020636

**Published:** 2023-01-13

**Authors:** Oskari Repo, Carlo M. Oranges, Pietro G. di Summa, Panu Uusalo, Mikael Anttinen, Salvatore Giordano

**Affiliations:** 1Department of Plastic and General Surgery, Turku University Hospital, University of Turku, 20500 Turku, Finland; 2Department of Plastic, Reconstructive, and Aesthetic Surgery, Geneva University Hospitals, Geneva University, 1205 Geneva, Switzerland; 3Department of Plastic, Reconstructive and Hand Surgery, University Hospital of Lausanne (CHUV), 1011 Lausanne, Switzerland; 4Perioperative Services, Intensive Care and Pain Medicine, Turku University Hospital, 20500 Turku, Finland; 5Department of Urology, Turku University Hospital, University of Turku, 20500 Turku, Finland

**Keywords:** abdominoplasty, Scarpa fascia preservation, massive weight loss, seroma, bariatric surgery

## Abstract

(1) Background: An increasing number of patients undergo bariatric surgery and seek body contouring surgery after massive weight loss (MWL). Abdominoplasty itself is associated with a high complication rate in these patients, particularly due to seroma formation. Scarpa fascia preservation (SFP) has been proven to be an efficient method of reducing seroma rates. We aimed to evaluate the possible benefits of SFP on massive weight loss patients comparatively. (2) Methods: This is a single-center retrospective comparative study encompassing 202 MWL patients operated between 2009 and 2019 at Turku University Hospital. Patients included in the study had a preoperative weight loss greater than 30 kg. Of them, 149 went through traditional abdominoplasty and 53 abdominoplasties with SFP. The primary outcome measure was seroma occurrence, while secondary outcomes included drainage amount, hospital stay, surgical site occurrence, and need for blood transfusion. (3) Results: The only statistically significant difference between groups on patients’ demographics was the sex ratio, favoring females in the control group (43:10, 81% vs. 130:19, 87%, *p* = 0.018). SFP significantly reduced seroma occurrence (9.4% vs. 26.2%, *p* = 0.011) and decreased mean drainage duration (3.7 ± 2.4 vs. 5.3 ± 3.2 days, *p* = 0.025). There was a trend towards lower drainage output (214.1 ± 162.2 mL vs. 341.9 ± 480.5 mL, *p* = 0.060) and fewer postoperative days on ward in the SFP group. Other complication incidences did not differ between the groups. The multivariable analysis did not show any significant factor for seroma formation or surgical site occurrence. (4) Conclusions: Preserving Scarpa fascia on MWL patients may result in decreased seroma occurrence and a shorter time to drain removal.

## 1. Introduction

Abdominoplasty is a surgical procedure that addresses the abdominal shape and contour, particularly after weight loss. By tightening the abdominal wall muscular fascia excess and removing the excess skin and fat, it is possible to restore and shape the abdominal contour [1]. The number of abdominoplasties performed has continually increased annually. In 2019, it was the fourth most executed aesthetic surgical procedure based on the annual statistics of the American Society for Aesthetic Plastic Surgery (ASAPS), with over 140,000 cases [2]. Conversely, with the increasing worldwide prevalence of morbid obesity (body mass index; BMI ≥ 35 kg/m^2^) and the increasing rate of bariatric surgery procedures, the number of abdominoplasties is expected to further rise [3]. Massive weight loss (MWL) may result in significant skin excess, especially in the abdomen, upper arms, and thighs, causing physical discomfort [4]. Body contouring surgery, including abdominoplasty, has been proven effective in improving the quality of life of patients with MWL [5]. In addition, post-bariatric surgery has also been associated with an increased ability to lose and maintain achieved weight [6,7]. However, abdominoplasty is often associated with relatively high complication rates in MWL patients, with an incidence between 20.2% and 32.6% [7,8,9]. Seroma formation is the most common complication of abdominoplasty, with an average incidence of 10% ranging from 5 to 43% [7,8,9,10]. Many different strategies have been proposed to mitigate this kind of complication, including combined liposuction [11,12], tissue glues [13,14], quilting sutures to minimize dead space [15,16], postoperative compression dressings [17], and the avoidance of electrocautery [18,19]. Seroma formation may be reduced by avoiding the classical fascial dissection plane and preserving Scarpa fascia and its sub-fascial fat or by combining other strategies such as fibrin glue or quilting sutures to the abovementioned procedures [7]. In a meta-analysis, abdominoplasty with Scarpa fascia preservation (SFP) was associated with a significantly decreased seroma formation, total drain output, the time required for suction drains, and shorter hospital stays. However, no decreasing incidence of other complications such as hematoma, bleeding, infection, or suture rupture was reported, while the aesthetic outcomes were equal to the classical abdominoplasty [20]. More recently, another meta-analysis could not demonstrate benefits in decreasing the complication rate using quilting sutures, drains, or SFP in different combinations [21]. However, most of the included patients in those meta-analyses had a relatively low BMI (<26 kg/m^2^), and the MWL population was underrepresented. A few small cohort studies have reported results and relatively controversial outcomes on SFP in MWL patients [22,23]. Therefore, further evaluation of this technique in MWL patients may clarify this issue.

The present study aimed to compare the outcomes of the SFP technique versus traditional abdominoplasty in MWL patients, particularly on seroma formation. Even though seroma formation is commonly associated with factors that are predictive of poor wound healing, we hypothesized that SFP may mitigate the incidence of postoperative seroma based on our clinical experience.

## 2. Materials and Methods

This was a retrospective study from a prospectively maintained database. It was approved by the local Institutional Review Board and conducted in accordance with the ethical principles of the World Medical Association Declaration of Helsinki. We included all MWL patients undergoing abdominoplasty at Turku University Hospital from 1 January 2009 until 31 December 2019, with a minimum follow-up of six months. Patients were identified from the hospital surgery registry. Before the analysis, all patient data remained anonymous. We followed the Strengthening the reporting of cohort studies surgery (STROCSS) guidelines for observational cohort studies [24].

Inclusion criteria were adequate chart data, preoperative weight loss of >30 kg (MWL), stable preoperative weight (±5 kg), and postoperative follow-up of at least six months. Patients were excluded from this study in cases which included a circumferential abdominoplasty, body lift or belt lipectomy, concomitant procedures, secondary abdominal procedures, revisions, and a follow-up shorter than six months, for the purposes of this study.

For the present study, patients were divided into two groups according to the type of abdominoplasty performed: SFP abdominoplasty (experimental group) and classic plane full abdominoplasty (control group). The indication for the technique used was at the discretion of the individual surgeon involved and based on the clinical circumstances over a 10-year period. Data collected from patients’ electronic medical records, including demographics, medical history, type of bariatric surgery, smoking history, weight loss, and surgical outcomes, were directly compared between the experimental and control groups. Having one or more of the following conditions was considered medical comorbidity: coronary artery disease, diabetes mellitus, hypertension, pulmonary disease, or renal disease. Patients taking any diabetic medication were graded as diabetic. Patients who reported smoking within four weeks preoperatively were considered active smokers.

The primary outcome measure was the postoperative seroma occurrence. Secondary outcomes were drainage amount, length of hospital stay, surgical site occurrence (SSO), and need for blood transfusion.

Seroma was considered as serous fluid collection or blood between the tissue layers and diagnosed clinically or with ultrasound imaging and requiring drainage in the office or operating room.

SSO was defined as any complication involving the abdominal area that underwent the procedure, and the severity was measured using the Clavien–Dindo classification [25]. Superficial wound infection was a surgical wound requiring antibiotics (cellulitis), and deep infection was an infection that required emergency drainage or hospitalization (abscess). Wound dehiscence was a skin wound separating >0.5 cm, including all the skin layers, leading to over two weeks of delayed healing or needing specialist dressing care. Fat necrosis was palpable firmness with a diameter >1 cm staying at least three months. Hematomas included a hemorrhage requiring blood cell transfusion (Clavien–Dindo grade II) or emergency exploration (Clavie–Dindo grade II).

All operations were performed under general anesthesia by the same local surgical team. SFP abdominoplasty or dual plane abdominoplasty was widely similar to classic full abdominoplasty [26]. An incision was made to the level of the symphysis pubis, 6–7 cm superior to the anterior vulva commissure, and the abdominal flap was elevated up to the subcostal margin. As an exception, abdominal flaps were elevated and avulsed on a plane of the Scarpa fascia on the infra-umbilical area, excluding a several centimeter-wide vertical line in the midline bilaterally, which reached the pre-muscular plane (Figure 1). Rectus fascia plication could be achieved using interrupted #1 polypropylene sutures. On the supra-umbilical and epigastric areas, flap elevation was carried out on the pre-muscular plane up to the costal margin.

Classic plane full abdominoplasty with umbilical transposition was performed on all patients, and rectus plication was accomplished in cases where a diastasis was larger than 5 cm at the surgeon’s discretion. Abdominal flap elevation started a few centimeters above the symphysis pubis from the “bikini line” up to the costal margin on the plane of the rectus fascia. After the excess of the abdominal flap was removed, the umbilicus transposed, hemostasis achieved, and two closed-suction drains were placed, skin closure was performed in two or three layers at the surgeon’s discretion. Some patients (26.6%) received pain pump catheters [27].

All patients were treated under general anesthesia according to the local protocol. Postoperative pain was alleviated with opioids and nonsteroidal anti-inflammatory drugs (NSAIDs). All patients received 40 mg of subcutaneous enoxaparin for thrombosis prophylaxis unless no contraindications were observed. Intravenous antibiotics were given as a single intravenous dose at the induction and continued according to the surgeon until the suction drains were removed. Surgical drains were removed when the output was less than 30 mL/day, and the maximum duration for surgical drains was 14 days. Follow-up was the patient’s last medical care contact at the study institution.

Continuous and categorical variables were reported as means (standard deviation [SD]) and counts (percentage), respectively. Normality assumptions were demonstrated with histograms, Skewness, Kurtosis, and Kolmogorov/Smirnov tests. Pearson’s chi-square test, Fisher’s exact test, and the t-test were used for univariate analysis, as appropriate, to compare the two study groups. Univariate and multivariable analyses were performed to assess the associations between seroma occurrence and SSO with patient and perioperative variables. In univariate analysis, perioperative factors with a *p*-value < 0.4 were included in a multivariable logistic regression model. Confidence intervals were set at 95%, and a two-sided *p*-value of <0.05 was considered statistically significant. All the analyses were conducted using SPSS version 28 (IBM SPSS Statistics, version 28, Armonk, NY, USA).

## 3. Results

A total of 202 consecutive patients undergoing abdominoplasty after MWL were included in the analysis after the eligibility assessment. Of them, 53 patients were included in the SFP group, while 149 patients were included in the control group (Figure 2). Patient demographics and study group comparisons are shown in Table 1. There was a statistically significant difference only in the sex ratio of the patients, where 87.2% of patients in the control group were female versus 81.1% in the SFP group (Table 1).

A significant difference was found in the mean drainage duration time favoring the SFP group (3.7 vs. 5.3 days, *p* = 0.025). A trend toward lower drainage output and shorter hospital stays was also noted (Table 2). The follow-up was significantly longer in the control group (33.9 vs. 60.4 days; *p* = 0.001). There were no other significant differences in the perioperative parameters, such as operation time, resection weight, or rectus plication occurrence (Table 2). Seroma occurrence was significantly reduced in the SFP group (9.4% vs. 26.2%, *p* = 0.011, Table 3). No other differences among the postoperative complication rates graded with the Clavien–Dindo classification were detected between the groups (Table 3), with no grade III or IV complications observed. The multivariable analysis did not identify any significant risk factor for postoperative seroma formation or surgical site occurrence (Table 4).

## 4. Discussion

Our findings support the hypothesis that SFP resulted in a significantly reduced drainage duration, lower drainage output, and shorter hospital stay in MWL patients compared to the traditional abdominoplasty. A continuously increasing number of patients seek body contouring surgery procedures after MWL caused by dietary changes and exercise or bariatric surgery. Body contouring operations have been shown to increase patients’ quality of life [5]; however, they carry a risk of complications. Of these procedures, abdominoplasty is the most common but includes the highest complication rate [28]. In order to achieve the best possible result benefitting patients’ quality of life, finding possible strategies to mitigate complications is essential. Studies on traditional abdominoplasty have reported an overall complication rate varying from 20.2% to 32.6% [7,8,9]. Seroma formation is the most common abdominoplasty complication with an incidence ranging from 5 to 43%, typically higher in MWL patients [7,8,9,10]. Our analysis also found a relatively high overall complication rate in both groups (50.9% vs. 58.4%), favoring the SFP group. Possible explanatory factors for these results may be related to the study population: all patients had a high preoperative BMI (near 30 kg/m^2^) and experienced MWL (mostly over 40 kg), and the proportion of smokers was relatively high (22.6–25.5%). These features have primarily been studied as risk factors for complications after abdominoplasty [9,10], inflicting higher frequency of seroma, longer drain duration, and greater drain output [9], whereas obesity and smoking are the leading risk factor for seroma formation [10]. Furthermore, according to a meta-analysis, the risk for complications in the post-bariatric population is 60–87% higher when compared to the population that achieved weight loss with dietary changes or exercise [29]. Our report did not identify any independent risk factors for SSO including seroma formation, probably due to the relatively small study population.

SFP in abdominoplasty was first introduced by Le Louarn [30]. After that, this strategy has been shown to be beneficial in several articles, resulting in reduced seroma formation and duration. Interestingly, most of these studies reported a decrease in total drain output [31,32,33,34,35,36], decreased time needed for drains [31,32,37], and a shorter hospital stay [37]. However, none of the studies reported a decreased seroma rate [37]. When ultrasounds were used to identify postoperative fluid collections, the incidence of seroma was similar [36]. Our outcomes are consistent with the previously published literature, despite our higher morbid population. Some surgeons consider the aesthetic results inferior when applying the SFP technique in abdominoplasty because untouching the deep fat layer above the rectus fascia may result in additional bulking in the lower abdomen. However, the aesthetic result seems to be equal to the traditional abdominoplasty [32].

To achieve reliable waist modification, different techniques have been published. L-shaped external oblique muscle plication, multidirectional abdominal wall plication, and advancement of the external oblique muscle flaps have been described [33]. However, plication of the external obliques demonstrates limited mobility, and the creation of widely undermined external oblique flaps is typically beyond the scope of the standard outpatient procedure. To overcome this issue, Scarpa’s fascia has been used to enhance waistline definition during abdominoplasty by some authors. Mossaad et al. [34] removed a full-thickness midline strip of subcutaneous tissue below the umbilicus down to the rectus sheath, with a tissue advancement through lipo-mobilization using standard liposuction techniques without undermining the above umbilicus other than the midline to define the waistline through medial directional pulling. More recently, Whiteman et al. [35] used Scarpa’s fascia flaps pulled in an infero-medial direction to improve the definition of the waistline from the upper abdominal flap. In our study, we did not assess the aesthetic outcomes concerning the superiority of SFP due to a lack of data.

A recent meta-analysis on the SFP in abdominoplasty reported decreased drain removal time, total drain volume, and hospital stay without benefits on hematoma, bleeding, infection, and wound dehiscence [20]. Most of the included studies had patients with a BMI close to normal, with only two studies on SFP abdominoplasty with MWL and/or bariatric patients, both including a relatively small cohort of 51 and 42 post-bariatric patients [22,23]. These studies detected a significantly lower drainage output and faster suction drain removal. One study also reported a reduced hospital stay [22], and the other did not find a benefit in the postoperative seroma formation rate identified using ultrasounds [23].

Recently, studies on abdominal wall anatomy focusing on the distribution of lymphatic vessels have been performed to clarify if the seroma formation is possibly caused by damage to the lymphatics and if SFP is beneficial because of the subsequent preservation of its underlying lymphatics [38,39]. An anatomical study on tissue samples from abdominoplasties found that around 17% of abdominal wall lymphatics were on the level of Scarpa fascia or beneath it and were most prevalent in the dermis [38]. Another study showed that lymphatic collectors of the abdominal wall run above the Scarpa fascia and concluded that saving lymphatics is not the mechanism behind fewer seroma rates [39]. Other considered explanations were reduced dead space [39] and better surface adhesion [32]. Moreover, a study on post-abdominoplasty seroma composition reported that it resembles inflammatory exudate in composition, and it is altered through days to exudate with some features similar to lymph. Because of the inflammatory nature of the exudate, the authors concluded that gentler tissue handling, including SFP, may be beneficial [40]. The mechanisms behind the benefits of SFP remain unclear, and our results were consistent with previous clinical studies despite the higher BMI and weight loss in the included patients. Our findings are valuable for high-risk patients without differences in operative time or a need for additional surgical equipment. Therefore, SFP seems to be a cost-saving strategy to reduce complications and extra costs involved with this procedure. From a technical point of view, detecting the level of Scarpa fascia might be more demanding; however, we believe that it is an easy technique to learn and be applied with relatively little practice.

The strengths of this study include its relatively large sample size, consistent surgical technique in the two groups, same surgical teams, long-term follow-up, and comparable groups in terms of comorbidities. The most predominant limitation of this study is its retrospective nature. Secondly, the lack of randomization might introduce a certain bias related to unmeasured confounding factors influencing the decision to preserve the Scarpa fascia for each abdominoplasty case. Abdominoplasties were performed by multiple surgeons over a 10-year period, with individual differences in their approaches and techniques during the time of the study that may have contributed to some variability and selection bias. Moreover, the presence of smokers may play a role in affecting the outcomes. Smokers were preoperatively instructed to quit; however, around a quarter of each group consisted smoking patients, which may have resulted in an increased complication rate and could be a source of further bias [10]. Finally, some minor complications may have been treated at primary healthcare facilities, and these possible occurrences have not been considered in our analysis. Further studies with larger cohorts are needed to ensure the exact benefits of SFP, particularly on MWL patients. Research on the nature of seroma formation and its related factors is also warranted.

## 5. Conclusions

Preserving Scarpa fascia on MWL patients may result in decreased seroma occurrence and earlier drain removal. Further larger studies are needed to verify our findings. Further evidence is warranted to assess the role of Scarpa fascia preservation with a larger number of participants and better comparable groups to examine which individuals benefit most from this practice.

## Figures and Tables

**Figure 1 jcm-12-00636-f001:**
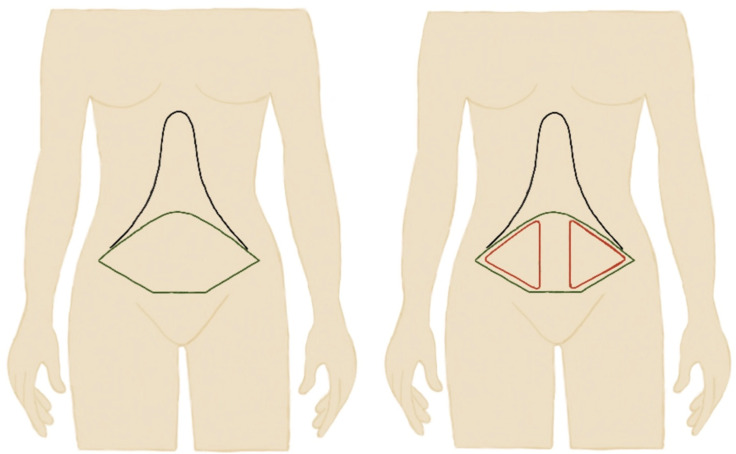
Schematic representation of classic full plane abdominoplasty versus Scarpa preservation abdominoplasty (SPF) where Scarpa fascia was preserved in the red areas.

**Figure 2 jcm-12-00636-f002:**
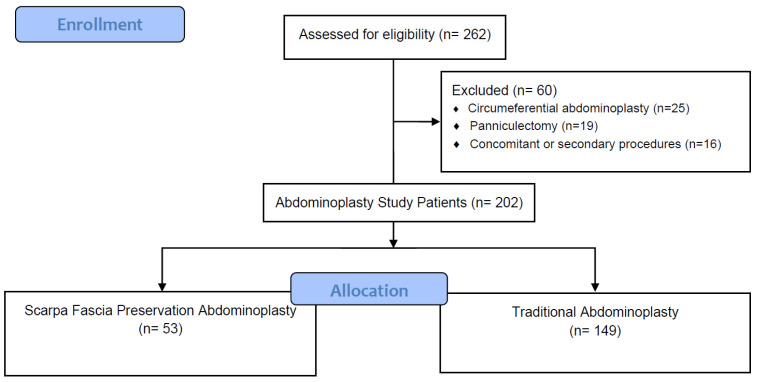
Flowchart of the study.

**Table 1 jcm-12-00636-t001:** Demographics of patients at the time of the study.

	Scarpa Fascia Preservation Group (n = 53)	Control Group (n = 149)	*p*-Value
Age (mean ± SD)	44.4 ± 10.79	44.0 ± 10.9	0.833
Sex ratio (F:M)	43:10	130:19	0.018
Bariatric Surgery (Y:N)	32:21	87:62	0.872
Mean BMI before abdominoplasty (kg/m^2^)	27.8 ± 2.9	28.3 ± 3.4	0.532
Mean weight loss (kg)	44.1 ± 20.0	39.3 ± 20.0	0.147
Any comorbidity	10 (18.9%)	41 (27.5%)	0.213
Diabetes	3 (5.7%)	21 (14.1%)	0.138
Smokers	12 (22.6%)	38 (25.5%)	0.678

**Table 2 jcm-12-00636-t002:** Comparison of peri-operative parameters in the two groups of patients.

	Scarpa Fascia Preservation Group (n = 53)	Control Group (n = 149)	*p*-Value
Operative time (min, mean ± SD)	152.2 ± 29.0	148.0 ± 48.1	0.566
Resection weight (g, mean ± SD)	1743.5 ± 750.6	1764.6 ± 1040.4	0.893
Rectus plication	41 (77.4%)	107 (71.8%)	0.433
Estimated blood loss (mL, mean ± SD)	206.1 ± 154.1	257.5 ± 182.1	0.072
Hospital stay (days, mean ± SD)	3.5 ± 1.5	4.0 ± 3.0	0.213
Total ward drainage (mL, mean ± SD)	214.1 ± 162.2	341.9 ± 480.5	0.060
Drainage first day (mL, mean ± SD) dx	39.9 ± 46.9	59.0 ± 91.3	0.152
Drainage first day (mL, mean ± SD) sx	45.6 ± 52.6	49.7 ± 64.8	0.687
Mean Drainage duration (days, mean ± SD)	3.7 ± 2.4	5.3 ± 3.2	0.025
Follow-up (months, mean ± SD)	33.9 ± 25.6	60.4 ± 52.5	0.001

**Table 3 jcm-12-00636-t003:** Postoperative complications at follow-up.

	Scarpa Fascia Preservation Group (n = 53)	Control Group (n = 149)	*p*-Value
Any complications (%)	27 (50.9%)	87 (58.4%)	0.348
Complications			
Clavien–Dindo grade I			
Superficial wound infection	7 (13.2%)	20 (13.4%)	0.588
Clavien–Dindo grade II			
Seroma	5 (9.4%)	39 (26.2%)	0.011
Blood Transfusion	5 (9.4%)	23 15.4%)	0.358
Clavien–Dindo grade III			
Hematoma	4 (7.5%)	21 (14.1%)	0.330
Deep wound infection	1 (1.9%)	13 (8.7%)	0.121
Wound dehiscence	6 (11.3%)	15 (10.1%)	0.797
Fat necrosis	1 (1.9%)	9 (6.0%)	0.460
Clavien–Dindo grade IV			
None	0 (0.0%)	0 (0.0%)	
Clavien–Dindo grade V			
None	0 (0.0%)	0 (0.0%)	

**Table 4 jcm-12-00636-t004:** Univariate and multivariable analyses of factors related to seroma formation and surgical site occurrence. Abbreviations: BMI, body mass index.

Seroma Formation	Hazard Ratio	95% Confidence Interval	*p*-Value
Univariate analysis			
Bariatric Surgery	3.9	0.1–15.5	0.207
Transfusion	3.7	1.6–8.4	0.002
Weight Loss	3.2	2.3–10.5	0.047
Hematoma	3.1	1.5–6.5	0.004
Infection	1.9	0.8–4.5	0.154
Comorbidity	1.7	0.8–3.4	0.138
Plication	1.1	0.9–1.2	0.101
Multivariable Analysis			
Comorbidity	3.2	0.6–2.3	0.613
Infection	2.0	0.9–4.2	0.073
Transfusion	1.4	0.4–5.1	0.595
Bariatric Surgery	1.3	0.7–2.4	0.450
Hematoma	1.2	0.4–4.0	0.772
Weight Loss	1.0	1.0–1.1	0.575
Plication	0.5	0.3–0.9	0.028
**Surgical Site Occurrence**			
Univariate Analysis			
Plication	1.2	1.0–1.5	0.075
Diabetes	1.0	0.5–1.9	0.858
Comorbidity	1.0	0.6–1.8	0.922
Smoking	1.2	0.7–2.2	0.473
Bariatric Surgery	1.7	1.0–2.7	0.047
Weight Loss	2.9	1.5–9.7	0.146
BMI	0.7	0.6–2.0	0.290
Multivariable Analysis			
Comorbidity	1.2	0.6–2.5	0.561
Bariatric Surgery	1.0	0.5–2.4	0.904
Weight Loss	1.0	1.0–1.1	0.058
BMI	1.0	0.9–1.1	0.455
Plication	0.6	0.3–1.3	0.178

## Data Availability

The data presented in this study are available on request from the corresponding author.

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
