# Peer review of "Scarpa Fascia Preservation to Reduce Seroma Rate on Massive Weight Loss Patients Undergoing Abdominoplasty: A Comparative Study"

_jcm, 2023, doi:10.3390/jcm12020636_

Round 1

Reviewer 1 Report

Dear author, thank you for the nice paper that is clinically relevant and interesting to the readers of JCM. 

Please elaborate on why the SFP was chosen by the operator significantly less freqeuently than classic abdominoplasty other than the technical challenge. Less operating time? 

There are operating techniques that use the scarpa fascia as advancement flaps for a lift of the suprapubic region and/or the waistline. It would be interesting to include a possible aesthetic evolution of the technique, as the mentioned additional bulk might be adressed by this. 

Please check the manuscript for typing errors, in the discussion the citations are at times not in square brackets. It would also benefit from the corrections of a natve english speaker, as there are some grammatical errors and false vocabulary. 

Author Response

Dear Editors and Reviewers,                                                                                                      

We wish to thank each of the Reviewers for their thoughtful and insightful comments on our manuscript entitled: “Scarpa Fascia Preservation to Reduce Seroma Rate on Massive Weight Loss Patients Undergoing Abdominoplasty: A Comparative Study”.

We have done our best to address each of the comments in our revised manuscript and feel that the changes that have been made have resulted in an even better manuscript that will be of use to all surgeons who are involved with massive weight loss patients undergoing abdominoplasty.

A point-by-point response to the Reviewers’ comments follows.  Changes are reported in red.

Comment: Dear author, thank you for the nice paper that is clinically relevant and interesting to the readers of JCM.

Response: We are very grateful for your comments. We have responded all comments and highlighted all changes made in our manuscript with red colour.

Comment: Please elaborate on why the SFP was chosen by the operator significantly less frequently than classic abdominoplasty other than the technical challenge. Less operating time?

Response: Thank you for your insightful comment. Abdominoplasties were performed by multiple surgeons over a 10-year period, with individual differences in their approaches and techniques that may have contributed to some variability. This has been now acknowledged in the limitation section. Please see pages 3 and 9.

Comment: There are operating techniques that use the Scarpa fascia as advancement flaps for a lift of the suprapubic region and/or the waistline. It would be interesting to include a possible aesthetic evolution of the technique, as the mentioned additional bulk might be addressed by this.

Response: Thank you for your interesting comment. Unfortunately, we have no data on this issue. However, we have now briefly discussed the evolution of the Scarpa fascia as advancement flap technique. Please see page 8.

Comment: Please check the manuscript for typing errors, in the discussion the citations are at times not in square brackets. It would also benefit from the corrections of a native English speaker, as there are some grammatical errors and false vocabulary.

Response: Thank you for noticing that. We have now revised the Discussion and its citations. A native English speaker has revised our manuscript. I hope the manuscript is now optimally phrased and free from typographical and grammatical errors.

Reviewer 2 Report

Thank you very much for the opportunity to review the article.
The number of bariatric surgeries and people with significant weight loss is increasing significantly every year.
The problem of acceptance of one's own body after significant weight loss is becoming important.

The article is written in a clear and transparent manner. However, the authors did not shy away from a few errors.

It should be presented in which years operations of what type were performed. Whether this isn't actually a comparison of two surgical methods but a study on the improvement of treatment results due to a change in the type of surgery.

The authors of the article indicated that revision after primary surgery disqualified from further study. It should be presented what percentage of patients were disqualified and the reason for this revision.

Author Response

Dear Editors and Reviewers,                                                                                                      

We wish to thank each of the Reviewers for their thoughtful and insightful comments on our manuscript entitled: “Scarpa Fascia Preservation to Reduce Seroma Rate on Massive Weight Loss Patients Undergoing Abdominoplasty: A Comparative Study”.

We have done our best to address each of the comments in our revised manuscript and feel that the changes that have been made have resulted in an even better manuscript that will be of use to all surgeons who are involved with massive weight loss patients undergoing abdominoplasty.

A point-by-point response to the Reviewers’ comments follows.  Changes are reported in red.

Comment: Thank you very much for the opportunity to review the article.
The number of bariatric surgeries and people with significant weight loss is increasing significantly every year.
The problem of acceptance of one's own body after significant weight loss is becoming important.
The article is written in a clear and transparent manner. However, the authors did not shy away from a few errors.

Response: We thank the Reviewer for His/Her comments and for the possibility to response to the comments. We agree with the Reviewer about some shortages in the manuscript and have now responded to all comments and highlighted all changes made in our manuscript with red colour. After revision, we ensured that the revised manuscript conforms to the journal style.

Comment: It should be presented in which years operations of what type were performed. Whether this isn't actually a comparison of two surgical methods but a study on the improvement of treatment results due to a change in the type of surgery.
Response: Thank you for your comment. We have now reported the years when the surgery was performed (Please see Abstract and Methods sections, pages 1 and 2).

Abdominoplasties were performed by multiple surgeons over a 10-year period, with individual differences in their approaches and techniques that may have contributed to some variability. Unfortunately, we are not able to detect how surgical technique implementation over the years have affected the outcomes. This has been now acknowledged in the limitation section. Please see pages 3 and 9.

Comment: The authors of the article indicated that revision after primary surgery disqualified from further study. It should be presented what percentage of patients were disqualified and the reason for this revision.

Response: Thank you for noticing that. The exclusion criteria for this study are explained on the methods section (Please see page 2). We have now added the flow-chart of this study (Please see Figure 2).